# Removal of Organic Micropollutants in Wastewater Treated by Activated Sludge and Constructed Wetlands: A Comparative Study

**Carolina Reyes Contreras** [1], **Daniela López** [1], **Ana M. Leiva** [1], **Carmen Domínguez** [2], **Josep M. Bayona** [2] **and Gladys Vidal** [1,*]

[1] Engineering and Environmental Biotechnology Group, Environmental Science Faculty and EULA-Chile Centre, Universidad de Concepción, 4070386 Concepción, Chile; carolinareyescontreras@gmail.com (C.R.C.); spdanielita@gmail.com (D.L.); anamarialeiva@udec.cl (A.M.L.)

[2] Department of Environmental Chemistry, Institute of Environmental Assessment and Water Research, IDAEA, c/Jordi Girona 18-26, E- 08034 Barcelona, Spain; cdfqst@cid.csic.es (C.D.); jbtqam@cid.csic.es (J.M.B.)

[*] Correspondence: glvidal@udec.cl; Tel.: +56-41-2204-067

**Abstract:** The aim of this study is to compare the removal of organic micropollutants (OMPs) in wastewater by activated sludge (AS) and constructed wetlands (CWs). This analysis was carried out in a wastewater treatment plant (WWTP) of a rural community where they implemented two technologies in parallel: AS and a pilot plant of horizontal subsurface flow (HSSF) constructed wetlands. In this case, these systems were fed by the same influent and the removal efficiencies of 14 OMPs, including analgesics/anti-inflammatories, anticonvulsants, stimulants, antifungals, fragrances, plasticizers, and transformation products, were evaluated in each system. Regarding the presence of OMPs in the wastewater, the concentrations of these compounds in the influent ranged from 0.16 to 7.75 µg/L. In general, the removal efficiencies achieved by the AS system were between 10%–95% higher than those values reported by HSSFs with values above 80% for naproxen, ibuprofen, diclofenac, caffeine, triclosan, methyl dihydrojasmonate, bisphenol-A, 2-hydroxyl ibuprofen, and carboxy ibuprofen ($p < 0.05$). This behavior can be related to the aerobic conditions that promote the AS system with oxidation-reduction potential (ORP) and dissolved oxygen (DO) values above −281 mV and 0.24 mg/L, respectively. However, the removal of galaxolide was greater in HSSF system than in AS with significant difference of 70% ($p < 0.05$). Despite these results, this study reveals that comparing both technologies, AS had the best removal performance of these OMPs studied.

**Keywords:** organic micropollutants; activated sludge; horizontal subsurface flow constructed wetlands

## 1. Introduction

In recent years, the studies related to the occurrence and fate of organic micropollutants (OMPs) have increased due to their potential toxicological effects on human health and the environment [1–3]. One of the principal sources of OMPs are wastewater treatment plants (WWTPs) that receive wastewater coming from domestic sewage, hospital sewage, and several human activities which contain a lot of different OMPs such as pharmaceutical and personal care products (PPCPs), pesticides, surfactants, industrial additives, and plasticizers [4,5].

The OMPs removal on WWTPs is generally influenced by the physicochemical properties of the compound; by the type of wastewater treatment technology (conventional and non-conventional) and by process-specific factors such as sludge retention time (SRT), temperature, and organic loading rates (OLR) [6]. Conventional treatment technologies are characterized by using mechanized technologies with high electrical energy requirements and the design, supervision, maintenance, and the general

cost of construction that would require highly skilled workers [7]. The most common conventional wastewater treatment is activated sludge (AS) that has not been specifically designed for removing OMPs [4]. However, several studies reported that some OMPs can be removed during AS by microbial processes (biodegradation) and adsorption onto sludge flocs. One of the advantages of this technology is that oxygen conditions are an important factor for the removal of some OMPs [8]. For ibuprofen, naproxen, galaxolide, and caffeine, the removal efficiencies reported in these systems varied between 60%–99%, 25%–98%, 70%–85%, and 77%–100%, respectively [9–12]. Despite these performances, some OMPs, such as carbamazepine, were not removed during AS achieving negative percentages [9,11].

On the other hand, non-conventional wastewater treatment technologies which are based on natural processes for removing pollutants with low cost and less complex operation and design, have been implemented for treating wastewater in rural areas [13]. Constructed wetlands (CWs) are cost-effective non-conventional technologies that have shown great capacity for OMPs removal in wastewater [14]. However, their implementation is only feasible in small communities due to the large surface needed per inhabitant [15]. The mechanisms for removing OMPs involved biotic processes such microbiological degradation and biofilm and plant uptake and physicochemical processes such as absorption into the support medium, photodegradation, and evaporation. Moreover, these processes depend on the CWs design that promote the conditions for carrying out [16]. According to their wastewater flow regime, CWs can be classified as surface flow (SF), horizontal subsurface flow (HSSF), and vertical subsurface flow (VSSF) [17]. In the case of HSSF, which promotes anaerobic conditions for removing compounds, the removal efficiencies of ibuprofen, naproxen, diclofenac, triclosan, galaxolide, tonalide, and methyl dihydrojasmonate vary between 11%–92%, 45%–88%, 41%–93%, 47%–87%, 42%–95%, 36%–64%, 64%–99%, and 67%–96%, respectively [16–19].

Different studies have focused on the effectiveness of conventional and non-conventional wastewater treatment for removing OMPs. Matamoros et al. [20] evaluated the removal efficiencies of 25 OMPs in four different conventional and non-conventional WWTPs which were AS, CWs, stabilization ponds (SPs), and rotating biological contactors (RBCs). This study showed that CWs had the lowest overall removal efficiency of all the evaluated technologies. Comparing AS and CW systems, the removal of caffeine, ibuprofen, and triclosan were 1.32, 8.9, and 2.4 times higher in the conventional wastewater treatment (AS). This difference can be explained by the redox conditions of these systems. It is reported that ibuprofen and caffeine were better removed under aerobic conditions that promote AS technologies [21,22]. The same behavior was observed by Qiang et al. [23] who investigated the removal efficiencies of six OMPs in 20 rural WWTPs comparing different treatment processes: AS, CWs, SPs, and micro-power biofilm reactors (MPs). In this study, AS achieved higher performances than CWs, SPs, and MP with removal efficiencies above 70%. For bisphenol-A, the removal efficiencies of CWs fluctuated between 30%–50%, whereas AS reported values that were two times higher than CWs. Moreover, Ahmed et al. [24] made a review that focused on the removal of OMPs by biological, chemical, and hybrid technologies in effluents from WWTPs. Comparing the removal efficiencies of different groups of OMPs (including PPCPs, endocrine disrupting compounds (EDCs), pesticides, and analgesics) between AS and CWs, similar performances were obtained for EDCs and PPCPs with percentages between 84%–90% and 82%–84%, respectively. However, for pesticides and analgesics, CWs reported higher removal efficiencies which were 20% and 15% above those achieved by AS. Some of these compounds are highly polar and AS with lower retention time limited the capacity of microorganisms to metabolize. Furthermore, Melvin and Leusch [25] performed a quantitative meta-analysis to compare the removal efficiencies of PPCPs achieved by AS, oxidation ditch (OD), membrane bioreactor (MBR), ponds, and CWs. Through the standardization of removal efficiencies, the results of this study demonstrated that AS and CWs systems achieved the poorest overall removal of OMPs with values of 60% and 62%, respectively.

Within these divided opinions, the aim of this study is to compare the performance of 15 OMPs which include analgesics/anti-inflammatories, anticonvulsants, stimulants, antibacterials/antifungals, fragrances, plasticizers, and transformation products in a conventional and non-conventional

wastewater technology that consists in AS and CWs, respectively. In this case, the AS technology is an extended aeration system whereas the CW consists in a pilot plant of HSSFs. Both systems treat wastewater of a rural community.

## 2. Materials and Methods

### 2.1. Description of AS and HSSF Systems

The two treatment technologies, AS and HSSF, selected for this study are located in Hualqui (36°59′26.93″ S and 72°56′47.23″ W), Biobío Region, Chile. Figure 1 shows a schematic diagram of AS and HSSF systems. These technologies were fed with the same influent. This raw wastewater was pre-treated (40 mm chamber bars and a sand trap), which was followed by secondary treatment (AS or HSSF). The AS began its operation in 2005 and it was designed to treat wastewater for a rural community of 20,000 inhabitants. This system consisted of an extended aeration AS with clarification. Finally, the effluent was chlorinated before being discharged into the receiving water body. Regarding operation parameters, the feeding rate, hydraulic retention time (HRT), and sludge retention time (SRT) were approximately 2211.84 $m^3$/d, 9 h, and 25 d, respectively.

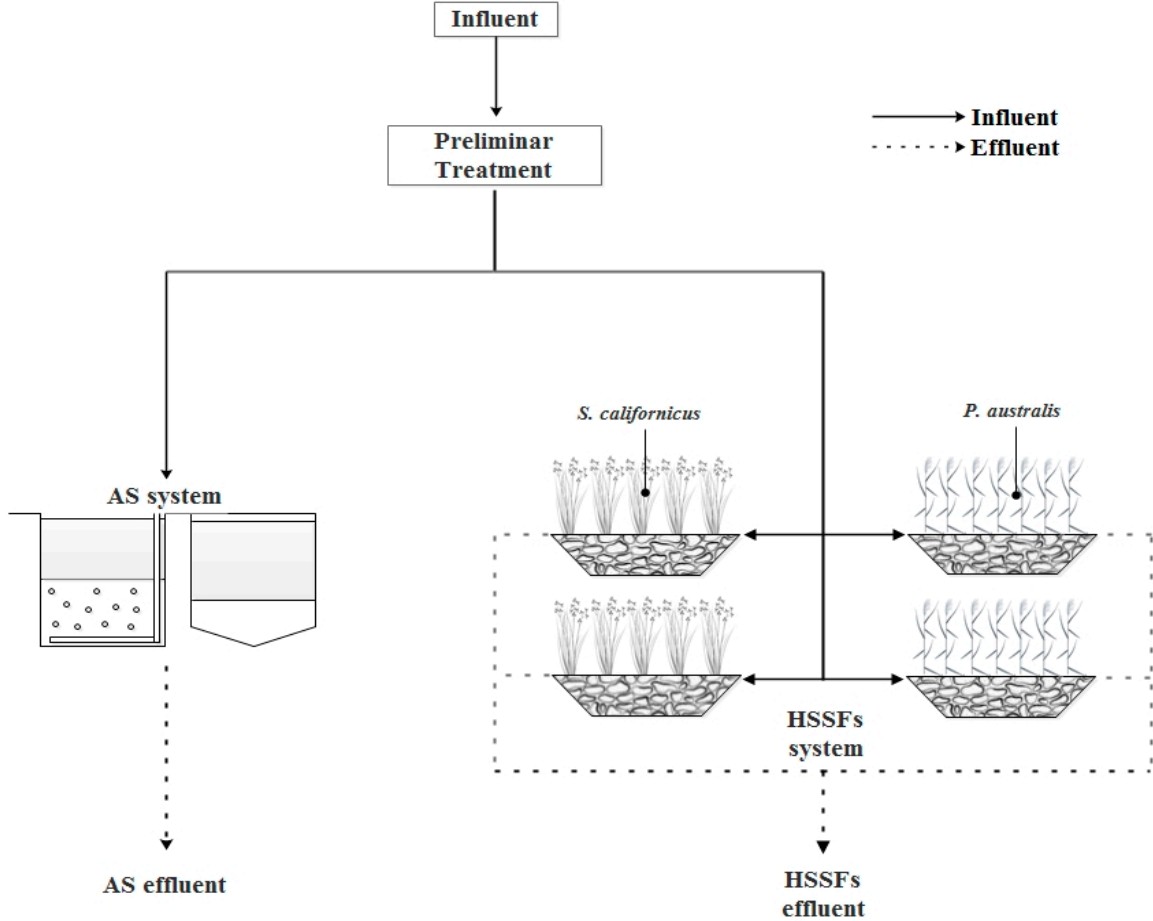

**Figure 1.** Schematic diagram of activated sludge (AS) and horizontal subsurface flow (HSSF) systems.

On the other hand, the CWs system was a pilot plant that consisted of a primary treatment (Imhoff tank and septic tank) followed by four parallel HSSFs units as a secondary treatment. Each HSSF had an area of 4.5 $m^2$ (length: 3 m and width: 1.5 m) and an average depth of 0.5 m as shown in Figure 2 [26]. The theorical volume of each unit was approximately 1.28 $m^3$. Regarding the support medium, HSSFs were filled with gravel which had a size of 19–25 mm and a porosity of 0.6%. The water level was maintained 0.10 m below the gravel surface to provide a water depth of 0.4 m. In each HSSF,

three samplings tubes were installed in each zone: Zone A (the inlet zone), 0.65 m from the inlet; Zone B (middle zone), 1.4 m from the entrance point; and Zone C (the outlet zone), 2.25 m from the inlet [27]. These systems were planted with *Phragmites australis* (*P. australis*) and *Schoenoplectus californicus* (*S. californicus*). At the time of the sampling campaign, the plant densities were $327 \pm 126$ stems/m$^2$ and $789 \pm 124$ stems/m$^2$ for *P. australis* and *S. californicus*, respectively [27]. Regarding the operational parameters, the feeding rate and HRT were 0.088 m$^3$/d and 5.1 d, respectively. As shown Figure 1, the four effluents generated by the different HSSFs were collected in a tank that represents the total effluent of the pilot plant of HSSFs.

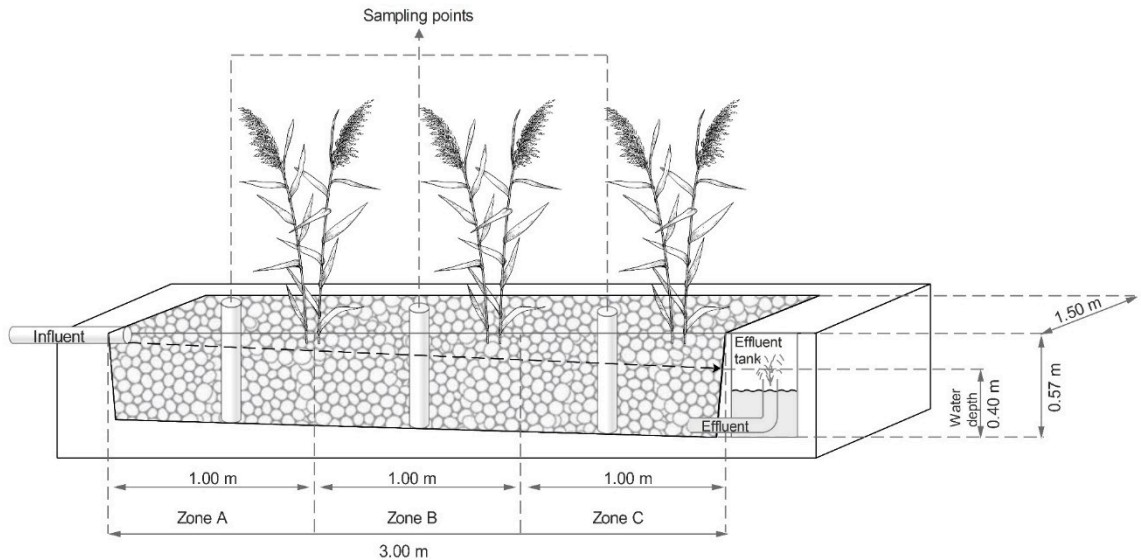

**Figure 2.** Sampling points of biomass on the gravel bed of HSSF system.

### 2.2. Sampling Strategy

Sampling was carried out in autumn (April 2014) and different types of samples were considered for this experiment: liquid and solid. The wastewater samples were collected twice a week during 1 month (4 weeks) from the influent and AS and HSSF effluents (6 samples per week). In this case, the influent was the same for both technologies (AS and HSSF), where samples were taken after pre-treatment (chamber bars and a sand trap). The effluent samples from the AS system were taken after the chlorination process. A total of 24 wastewater samples were obtained. The samples were collected in 1 L amber glass bottles and transported under refrigeration to the laboratory, where they were stored at 4 °C and processed within 24 h.

For solids samples, biosolids from AS and gravel bed biosolids from HSSF were obtained in the final sampling. The biosolids from AS were collected in the final stage after dehydration obtaining composite samples for stabilized sludge and they were analyzed in triplicate. In the case of gravel bed biosolids from HSSF, samples were obtained for three zones of the HSSF beds: Zones A, B, and C (Figure 2). All samples were frozen at −20 °C and freeze-dried before analysis.

### 2.3. Analytical Methods

#### 2.3.1. Physicochemical and Water Quality Parameters

The in situ parameters, such pH, temperature, electrical conductivity (EC), dissolved oxygen (DO), and oxidation-reduction potential (ORP), were measured in situ with all the samples using a multiparametric OAKTON-PC650 (Eutech Instruments; Singapore city Singapore). Conventional wastewater parameters, including chemical oxygen demand (COD), biological oxygen demand (BOD$_5$), total suspended solids (TSS), volatile suspended solids (VSS), ammoniacal nitrogen (NH$_4^+$-N), and orthophosphate (PO$_4^{3-}$-P), were analyzed following standard methods [28]. Total organic carbon

(TOC) was measured using a TOC-L CPH analyzer (Shimadzu; Tokyo Japan) equipped with an ASI-L autosampler (Shimadzu; Tokyo Japan).

### 2.3.2. Determination of OMPs and Transformation Products by Biodegradation

The OMPs selected included analgesics/anti-inflammatories (naproxen, ibuprofen, and diclofenac), anticonvulsants (carbamazepine), stimulants (caffeine), antibacterials/antifungals (triclosan), fragrances (methyl dihydrojasmonate, tonalide and galaxolide), plasticizers (bisphenol-A), and transformation products (1-hydroxy ibuprofen, 2-hydroxy ibuprofen, carboxy ibuprofen, ibuprofen amide, and 4-hydroxy diclofenac).

For wastewater samples: The OMPs and their transformation products by biodegradation in the wastewater samples were analyzed following a previously described methodology [22]. Briefly, all the samples were filtered through a glass fiber filter with a pore size of 0.7 μm and then acidified to pH 3 using concentrated hydrochloric acid. Sample volumes of 100 mL for the influent and 250 mL for the effluent were then spiked with 0.5 μg of a surrogate standard mix (i.e., fenoprop for the acidic compounds, 2,2'-dinitrobiphenyl for the fragrances, and 10,11-dihydrocarbamazepine for the neutral compounds). Polymeric solid-phase extraction cartridges were conditioned with 5 mL of n-hexane, 5 mL of ethyl acetate, 5 mL of methanol, and 5 mL of Milli Q water (pH 3). The spiked samples were percolated through the cartridges, allowed to dry for 30 min, and eluted with 10 mL of hexane/ethyl acetate (1:1). The extract was evaporated to ca. 20 μL under a gentle nitrogen stream, and 125 ng triphenylamine were added as an internal standard. The vial was then reconstituted with ethyl acetate to a final volume of 300 μL.

Methylation of the carboxylic acidic group was performed in a hot Gas Chromatography (GC) injector (280 °C) by adding 10 μL of trimethyl sulfonium hydroxide to a 50 μL sample before injection. The derivatized samples were injected into a Bruker 320 MS GC-MS/MS in the electron impact mode (70 eV ionization energy) and fitted with a 20 m × 0.18 mm i.d. coated with 0.18 μm film thickness TRB-5 MS column purchased from Teknokroma (Sant Cugat del Vallès, Spain). Helium was used as a carrier gas (99.9995% purity) at a constant linear average speed of 40 cm s$^{-1}$. The oven temperature was held at 60 °C for 3.5 min$^{-1}$, then increased at 30 °C min$^{-1}$ to 150 °C, and finally increased at 8 °C min$^{-1}$ to 320 °C, with the final temperature held for 10 min. A sample volume of 2 μL was injected in the splitless mode at an injector temperature of 280 °C, and the purge valve was activated 50 s after the injection. The transfer line and ion source were set to 250 °C and 200 °C, respectively. The target compounds were analyzed in the multiple reaction monitoring (MRM) mode, which monitored two transitions for each compound. Physical chemical properties of the target OMPs are shown in Table 1.

**Table 1.** Physicochemical properties of the target organic micropollutants (OMPs) (pK$_a$, Log K$_{ow}$, Log K$_{oc}$, and solubility values). This data was obtained from EPI suite v4.00 and Log D$_{ow}$ were calculated at pH 8.3.

| OMPs | Uses | pK$_a$ | Log K$_{ow}$ | Log K$_{oc}$ | Log D$_{ow}$ | Water Solubility at 25 °C (mg/L) | Molecular Structure |
|---|---|---|---|---|---|---|---|
| Naproxen | Analgesic/ anti-inflammatory | 4.15 | 3.10 | 1.97 | 0.25 | 44.07 |  |
| Ibuprofen | | 4.91 | 3.79 | 2.35 | 1.70 | 57.97 |  |
| Diclofenac | | 4.15 | 4.02 | 2.61 | 1.17 | 10.89 |  |
| Carbamazepine | Anticonvulsant | n.a. | 2.25 | 2.28 | n.a. | 30.48 |  |
| Caffeine | Stimulant | −0.07 | 0.16 | 0.98 | −6.91 | 11,308 |  |

**Table 1.** *Cont.*

| OMPs | Uses | pK$_a$ | Log K$_{ow}$ | Log K$_{oc}$ | Log D$_{ow}$ | Water Solubility at 25 °C (mg/L) | Molecular Structure |
|------|------|--------|--------------|--------------|--------------|----------------------------------|---------------------|
| Triclosan | Antibacterial/ antifungal | 7.90 | 4.66 | 3.93 | 3.66 | 9.30 | |
| Methyl dihydrojasmonate | Fragrances | n.a. | 2.98 | 2.70 | n.a. | 154.88 | |
| Tonalide | | n.a. | 6.35 | 4.72 | n.a. | 0.37 | |
| Galaxolide | | n.a. | 6.26 | 4.10 | n.a. | 0.20 | |
| Bisphenol-A | Manufacture of plastics and epoxy resins | 8.73 | 3.64 | 3.10 | 3.63 | 146.15 | |

**Table 1.** *Cont.*

| OMPs | Uses | pK$_a$ | Log K$_{ow}$ | Log K$_{oc}$ | Log D$_{ow}$ | Water Solubility at 25 °C (mg/L) | Molecular Structure |
|------|------|--------|--------------|--------------|--------------|----------------------------------|---------------------|
| 1-Hydroxy ibuprofen | Ibuprofen transformation products | 4.55 | 2.25 | 0.99 | −3.39 | 3192 | |
| 2-Hydroxy ibuprofen | | 4.63 | 2.29 | 1.01 | −3.31 | 2974 | |
| Carboxy-ibuprofen | | 3.97 | 1.97 | 1.25 | −4.03 | 1453 | |
| Ibuprofen amide | | n.a. | −2.75 | −1.03 | n.a. | 8083 | |

Notes: n.a.: does not apply.

The linearity range was from 0.006 to 3 µg $L^{-1}$. The surrogate recoveries for the acidic, neutral compounds, and fragrances were 88 ± 3%, 81 ± 5%, and 86 ± 2%, respectively. The correlation coefficients (r2) of the calibration curves always exceeded 0.999.

For biosolids samples: The glass fiber filters containing the Suspended Solids (SS), biosolids, and gravel bed biomass was previously freeze-dried and spiked with 50 ng of the standard surrogate mix (i.e., fenoprop for acidic compounds, 2,2′-dinitrobiphenyl for fragrances, and 10,11-dihydrocarbamazepine for neutral compounds). They were then allowed to stand for 12 h at 4 °C to achieve equilibrium. Samples were extracted by sonication with 7 mL hexane-acetone (1:1, v/v) in three 10-min intervals. Extracts were recovered by liquid–solid phase separation, and the liquid volume was reduced under a gentle stream of nitrogen. The sample cleanup was performed with a Florisil column deactivated at 1% (3 g) and eluted with 5 mL of ethyl acetate. The volume was then reduced under a gentle stream of nitrogen, and 125 ng of triphenylamine was added as an internal standard, yielding a final volume of 300 µL. The cleaned samples were injected into a Bruker 320-MS GC-MS/MS.

The surrogate recoveries for the acidic and neutral compounds and fragrances were 81 ± 4 %, 87 ± 6%, and 82 ± 8%, respectively. The correlation coefficients (r2) of the calibration curves always exceeded 0.999. Table 2 shows the detection and quantification limits.

**Table 2.** Minimum, maximum (in parentheses), and average concentrations of the physicochemical and water quality parameters in the influent and effluent of AS and HSSFs (*n* = 8).

| Parameters | Influent | Effluents | |
|---|---|---|---|
| | | **AS** | **HSSFs** |
| pH | (7.95–8.81) | (6.89–7.17) | (7.05–7.75) |
| Temperature (°C) | 17 (15–20) | 17 (15–20) | 16 (15–20) |
| EC (mS/cm) | 3799 (2947–4534) | 1962 (1574–2293) | 3396 (2481–4293) |
| ORP (mV) | −229 (−291−−154) | 369 (300–457) | −281 (−241–(−341)) |
| DO (mg/L) | 0.53 (0.42–0.65) | 5.53 (4.25–7.32) | 0.24 (0.07–0.84) |
| COD (mg/L) | 274 (115–465) | 10.9 (4.6–18.5) | 74.4 (31.2–126.2) |
| $BOD_5$ (mg/L) | 184 (48–372) | 0.4 (0.1–0.8) | 77.5 (20.2–156.8) |
| $PO_4^{3-}$- P (mg/L) | 10 (5–15) | 1.8 (0.9–2.7) | 16 (14–18) |
| $NH_4^+$- N (mg/L) | 86 (31–190) | 21.9 (7.9–48.3) | 53.4 (19.2–118.0) |
| TC (mg/L) | 189 (162–210) | 50 (39–67) | 114 (97–137) |
| TOC (mg/L) | 83 (56–105) | 10 (7.66–15) | 20 (15–23) |
| TSS (mg/L) | 255 (76–565) | 5.6 (1.7–12.4) | 67.0 (20.0–148.5) |
| VSS (mg/L) | 193 (33–513) | 3.5 (0.6–9.4) | 37.6 (6.4–100.0) |

Notes: EC: electrical conductivity; ORP: oxidation-reduction potential; DO: dissolved oxygen; COD: chemical oxygen demand; $BOD_5$: biological oxygen demand; $PO_4^{3-}$P: orthophosphate; $NH_4^+$-N: ammoniacal nitrogen; TC: total carbon; TOC: total organic carbon; TSS: total suspended solids; VSS: volatile suspended solids.

### 2.3.3. Extractable Organic Matter

The extractable organic matter (EOM) contained in the gravel was determined by gravimetry. Approximately 1 mL of extract that had been previously obtained was added to a Petri dish and placed in an oven for 1 h at 105 °C, at which point a constant weight was reached.

### 2.4. Statistics Analyses

The experimental results were statistically evaluated using STATA®software, release 9. In this study, these analyses were performed on the data corresponding to concentrations achieved in the influent and the effluents from AS and HSSFs. Statistical significance differences of OMP concentration after each treatment (AS and HSSFs) compared to influent samples were tested by Wilcoxon signed-rank test (non-parametric test), since data do not exhibit a symmetric or normal distribution against a null hypothesis of no difference. A significance level of *p* = 0.05 was considered statistically significant.

## 3. Results and Discussion

*3.1. Conventional Water Quality Parameters of Influent and Effluents from AS and HSSFs*

Table 2 shows the average concentrations of water quality parameters of influent and effluents from AS and HSSFs. Regarding the in situ parameters such pH and temperature, the values varied between 7.0–8.5 and 16–17 °C in the influent and effluents evaluated without significant differences between AS and HSSF effluents. However, a significant difference between AS and HSSF effluents was reported for EC values. In this case, HSSF effluent had EC 1.7 times higher than the AS effluent. Evapotranspiration process that takes place in the HSSF system was responsible for the salinity increment of the effluent [29].

According to the classification established by Henze et al. [30], the influent used in this study can be classified such as "diluted" wastewater due to the concentration of COD, BOD$_5$, and TOC of 274, 184, and 83 mg/L, respectively. These values are similar to those reported by Leiva et al. [31] and Burgos et al. [32]. However, the NH$_4^+$-N concentration of the influent corresponds to a "concentrated" wastewater with values above 80 mg/L [30]. These behaviors are possibly related to the water use and the economic activity of the rural community focused on the agriculture. Figure 3 summarizes the mean removal efficiencies of water quality parameters for AS and HSSFs. In general, the AS system achieved higher performance of pollutants with average values between 74%–99% than the HSSF system which reported values between 35%–80%. For organic matter removals that included parameters such TSS, VSS, COD, BOD$_5$, and TOC, AS reported average percentages above 85%. In the case of HSSFs, the higher performance was 77% which was achieved by the removal of TSS and VSS. The same tendency was observed in Matamoros et al. [20] where the removal of COD on the AS system was 40% higher than on HSSF system in warm season. Moreover, they affirmed that AS had the best removal performance for all the studied water quality parameters compared to other technologies such CWs, RBC, and SP due to the forced aeration process that takes place in AS. This characteristic also influenced the removal of NH$_4^+$-N in AS which had a mean performance of 74.6%. As shown Table 2, the DO and ORP concentrations of AS proved the aerobic conditions of this system with values that ranged between 300–457 mV and 4.25–7.32 mg/L, respectively. With these conditions (DO > 1.5 mg/L), nitrification takes place in the AS system which is considered the main pathway for NH$_4^+$-N [33,34]. On the contrary, it is known that HSSFs promote anaerobic conditions [30]. In this system, the DO and ORP values fluctuated between 0.07–0.84 mg/L and (−241)–(−341) mV, respectively.

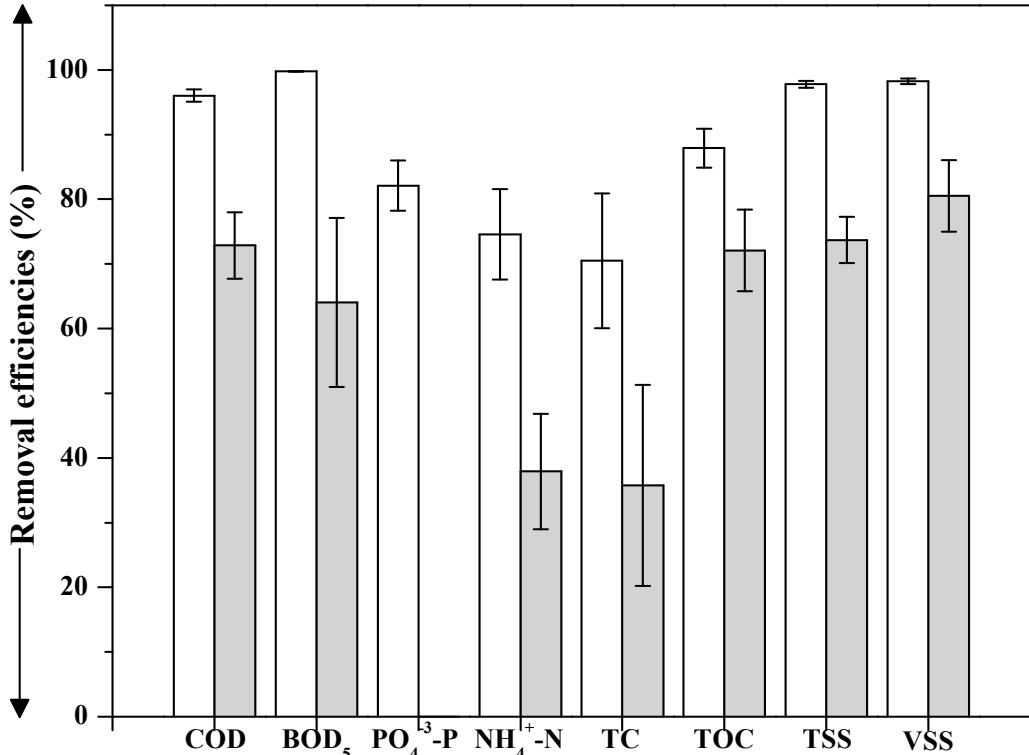

**Figure 3.** Removal efficiencies (bar chart) of quality water parameters for AS ( ▢ ) and HSSFs ( ▨ ). Error bars represent standard deviations.

### 3.2. Occurrence of OMPs in the Influent

The OMPs included in this study were selected based on the concentrations and frequencies of detection reported by several authors for surface water and WWTP effluents [5,35]. Table 3 shows mean, median, and range of concentrations ($n$ = 4–5) for all the OMP compounds studied in the influent and effluents of the evaluated systems. The OMPs detected in the influent were naproxen, ibuprofen, diclofenac, carbamazepine, caffeine, triclosan, methyl dihydrojasmonate, galaxolide, and bisphenol-A. The concentrations of micropollutants in the influent ranged from 0.16 to 7.75 µg/L (Table 2).

Ibuprofen, caffeine, and methyl dihydrojasmonate were the most abundant compounds in influent with concentrations of 7.7, 7.3, and 3.5 µg/L, respectively. These values are in the ranges found by Luo et al. [5] who evaluated the occurrence of OMPs in the influent of WWTPs. In this study, the concentration of ibuprofen and caffeine varied between 0.004–603 µg/L and 0.22–209 µg/L, respectively. However, the concentration of methyl dihydrojasmonate found in this study was approximately 50% lower than those reported by Matamoros et al. [18] and Hijosa-Valsero et al. [16] in influent of CWs. These values varied between 7.0–8.0 µg/L. These results are expected due to their physicochemical properties. These compounds (ibuprofen, caffeine, and methyl dihydrojasmonate) have hydrophilic characteristics with the octanol-water partition coefficient (LogK$_{ow}$) of 3.79, 0.16, and 2.98, respectively as shown Table 1.

Carbamazepine, triclosan, and bisphenol-A were found in low concentrations with values of 0.22, 0.20, and 0.16 µg/L, respectively. Comparing with literature, carbamazepine and triclosan are similar to those concentration reported in the literature. These values found in wastewater effluent varied between 1–1194 ng/L and 0.03–23.90 µg/L, respectively [5,36]. Moreover, the low concentration of bisphenol-A is associated with its characteristics of moderate solubility (water solubility: 146.15 mg/L) (Table 1) and because it can be partially adsorbed by sediments (Koc: 1245 L/kg).

**Table 3.** Average concentrations (μg/L) and standard deviations of OMPs and their transformation products evaluated in the influent and effluents of AS and HSSFs, respectively (*n* = 8).

| OMPs (μg/L) | Influent | | | Effluents | | | | | |
|---|---|---|---|---|---|---|---|---|---|
| | | | | AS | | | HSSFs | | |
| | Mean | Median | Range | Mean | Median | Range | Mean | Median | Range |
| Naproxen | 2.29 ± 0.72 | 2.43 | 1.94–3.69 | 0.010 ± 0.001 | 0.01 | 0.005–0.01 | 1.01 ± 0.26 | 1.14 | 0.76–1.39 |
| Ibuprofen | 7.75 ± 0.96 | 8.03 | 6.11–8.61 | 0.040 ± 0.003 | 0.03 | 0.03–0.04 | 2.62 ± 0.30 | 2.75 | 2.40–3.08 |
| Diclofenac | 1.03 ± 0.30 | 0.91 | 0.76–1.51 | 0.20 ± 0.04 | 0.19 | 0.16–0.26 | 0.28 ± 0.06 | 0.29 | 0.25–0.39 |
| Carbamazepine | 0.22 ± 0.05 | 0.21 | 0.19–0.31 | 0.16 ± 0.03 | 0.17 | 0.12–0.18 | 0.16 ± 0.02 | 0.15 | 0.15–0.20 |
| Caffeine | 7.33 ± 1.29 | 8.15 | 5.27–8.20 | 0.12 ± 0.01 | 0.12 | 0.10–0.13 | 0.57 ± 0.15 | 0.59 | 0.37–0.69 |
| Triclosan | 0.20 ± 0.06 | 0.19 | 0.15–0.30 | 0.020 ± 0.003 | 0.02 | 0.018–0.025 | 0.11 ± 0.04 | 0.11 | 0.06–0.14 |
| Methyl dihydrojasmonate | 3.49 ± 1.54 | 3.58 | 1.04–4.89 | 0.18 ± 0.01 | 0.18 | 0.16–0.19 | 0.53 ± 0.10 | 0.55 | 0.34–0.58 |
| Tonalide | <LOD | <LOD | <LOD | <LOD | <LOD | <LOD | <LOD | <LOD | <LOD |
| Galaxolide | 1.41 ± 0.31 | 1.21 | 0.75–1.58 | 1.15 ± 0.18 | 1.11 | 0.97–1.39 | 0.73 ± 0.13 | 0.76 | 0.57–0.89 |
| Bisphenol-A | 0.16 ± 0.05 | 0.13 | 0.12–0.23 | 0.020 ± 0.002 | 0.018 | 0.016–0.02 | 0.07 ± 0.02 | 0.06 | 0.05–0.11 |
| 1-hydroxy ibuprofen | 8.32 ± 1.32 | 8.22 | 7.17–10.53 | <LOD | <LOD | <LOD | 3.26 ± 0.35 | 3.11 | 3.02–3.66 |
| 2-hydroxy ibuprofen | 29.24 ±9.87 | 29.86 | 26.38–50.45 | 0.78 ± 0.29 | 0.72 | 0.48–1.18 | 28 ± 2 | 27.77 | 24.83–29.55 |
| Carboxy-ibuprofen | 53.58 ± 7.94 | 55.65 | 42.50–60.52 | 0.21 ± 0.06 | 0.20 | 0.14–0.28 | 41.40 ± 2.34 | 40.35 | 38.98–44.21 |
| Ibuprofen amide | 0.56 ± 0.14 | 0.43 | 0.43–0.70 | <LOD | <LOD | <LOD | n.d. | n.d | n.d |

Notes: LOD: limit of detection; n.d.: not detected.

Regarding transformation products, the compounds detected were carboxy-ibuprofen, 1-hydroxy-ibuprofen, 2-hydroxy-ibuprofen, and ibuprofen amide with concentrations of 54 μg/L, 8.32 μg/L, 29 μg/L, and 0.56 μg/L, respectively. The high concentrations of transformation products from ibuprofen are due to its widespread use in pharmaceuticals. Studies reported that carboxy-ibuprofen, 1-hydroxy ibuprofen, and 2-hydroxy ibuprofen are the main ibuprofen metabolites generated in biotransformation and biodegradation in the human body [30,37]. It is estimated that 15% of ibuprofen that is consumed is excreted as the parent compound, 26% is excreted as hydroxy ibuprofen, and 43% as carboxy-ibuprofen [38]. However, recent studies have reported that these OMPs are also generated in the biodegradation that occurs in WWTPs [39]. In this study, carboxy-ibuprofen and 2-hydroxy-ibuprofen were found to have higher concentrations in the influent than those reported by Hijosa-Valsero et al. [16]. In this case, the concentrations of these compounds were 7.84 and 12.1 μg/L, respectively.

On the other hand, Table 4 shows the average concentrations for the OMPs detected in the particulate phase. Carbamazepine, triclosan, tonalide, galaxolide, and bisphenol-A were detected. Galaxolide was the most abundant compound with a concentration of 0.55 μg/L, whereas carbamazepine, triclosan, and bisphenol-A achieved low concentrations ranging from 0.02 to 0.14 μg/L. This result was expected because galaxolide had a log Kow above 6.0 which indicated that this compound is hydrophobic and tends to be adsorbed by the solid phase [40].

**Table 4.** Average concentrations (μg/L) and standard deviations of selected OMPs in the suspended solids of influent samples and effluents of AS and HSSFs (*n* = 8).

| OMPs (μg/L) | Influent | Effluents | |
|---|---|---|---|
| | | AS | HSSFs |
| Carbamazepine | 0.14 ± 0.05 | 0.05 ± 0.01 | 0.12 ± 0.03 |
| Triclosan | ± 0.01 | 0.003 ± 0.001 | <LOD |
| Tonalide | <LOD | <LOD | <LOD |
| Galaxolide | 0.55 ± 0.14 | 0.13 ± 0.04 | 0.37 ± 0.13 |
| Bisphenol–A | 0.02 ± 0.01 | 0.006 ± 0.001 | 0.010 ± 0.003 |

Notes: LOD: limit of detection.

### 3.3. Ocurrence of OMPs in Biosolids

Table 5 shows the EOM and OMPs in the AS biosolids and the HSSF gravel bed biomass. For the AS biosolids, the EOM values achieved were 1912 µg/kg·dry weight (DW). In the case of HSSF system, the EOM values were 246, 225, and 101 µg/kg·DW for Zones A, B, and C, respectively. These results were lower than those reported by Hijosa-Valsero et al. [16] which presented values between 50–160 mg/kg·DW. Regarding the OMPs, the highest concentrations in the AS biosolids were galaxolide and 1-hydroxy ibuprofen with concentrations of 0.37 µg/kg·DW and 0.11 µg/kg·DW, respectively. The same tendency was observed in HSSF gravel bed biomass where galaxolide and 1-hydroxy ibuprofen had 0.001–0.020 µg/kg·DW and 0.006–0.009 µg/kg·DW, respectively, for Zones A, B, and C. For galaxolide and 2-hydroxy ibuprofen, the concentrations decreased by 95% and 50%, respectively along the HSSFs bed (from Zone A to Zone C). These difference along the longitudinal profile of HSSFs revealed that the highest removal of these compounds takes place within the first meter of the inflow zone (Zone A). This tendency was observed by Carranza-Diaz et al. [41] who evaluated the evolution of concentrations of three selected OMPs along the CWs. The concentration of caffeine decreased in 75% within the first 1.2 m of the inflow zone. Moreover, Vymazal and Kröpfelová [42] reported that removal of contaminants takes place within the first meters of CW beds.

**Table 5.** Average concentrations and standard deviations of extractable organic matter (EOM: µg/kg), OMPs and their transformation products (µg/kg) in the AS and HSSFs biosolids (*n* = 3).

| Parameters | AS | HSSFs | | |
|---|---|---|---|---|
| | | Zone A | Zone B | Zone C |
| EOM (µg/kg) | 1912 ± 436 | 246 ± 16 | 225 ± 17 | 101 ± 5 |
| Triclosan (µg/kg) | 0.010 ± 0.002 | 0.0030 ± 0.0005 | <LOD | <LOD |
| Galaxolide (µg/kg) | 0.37 ± 0.03 | 0.020 ± 0.003 | 0.0070 ± 0.0002 | 0.0010 ± 0.0001 |
| Bisphenol-A (µg/kg) | 0.0010 ± 0.0001 | <LOD | <LOD | <LOD |
| 1-Hydroxy ibuprofen (µg/kg) | 0.110 ± 0.006 | 0.009 ± 0.001 | 0.006 ± 0.001 | <LOD |
| 2-Hydroxy ibuprofen (µg/kg) | 0.030 ± 0.004 | 0.0020 ± 0.0003 | 0.0010 ± 0.0002 | <LOD |

Notes: LOD: limit of detection; EOM: extractable organic matter.

### 3.4. Removal Efficiencies of OMPs

Figure 4 shows the removal efficiencies of the OMPs in the AS and HSSF system. The removal efficiencies of OMPs and their transformation products was determined by their physicochemical properties and the operating parameters of the treatment systems.

Regarding naproxen and ibuprofen removals by both systems (AS and HSSFs), significant differences were observed ($p < 0.05$). In the case of the AS system, these compounds were efficiently removed with performances above 95%. However, they were partially removed by the HSSF system with values that varied between 55%–68%. Matamoros et al. [20] observed the same tendency with differences of 50% and 88% between AS and HSSFs for naproxen and ibuprofen, respectively. This behavior was expected because naproxen and ibuprofen are eliminated mainly under aerobic conditions or positive ORP [43]. In this study, the ORP values varied between 300–457 mV in AS systems. Under anaerobic conditions which promote HSSF system (ORP: (−241)–(−341) mV and DO: 0.07–0.84 mg/L), the removal of naproxen and ibuprofen is variable. Alvarino et al. [44] evaluated the removal efficiencies of naproxen and ibuprofen in an anaerobic reactor. As reported in this study, naproxen can be removed under both aerobic and anaerobic conditions, whereas ibuprofen is recalcitrant to biodegradation under anaerobic conditions achieving performance below 15%. Moreover, it was reported that the removal of ibuprofen and naproxen in HSSF system varies seasonally. In this context, Hijosa-Valsero et al. [45] reported removal efficiencies below 45% for both compounds in winter, whereas in summer, over 90% was removed.

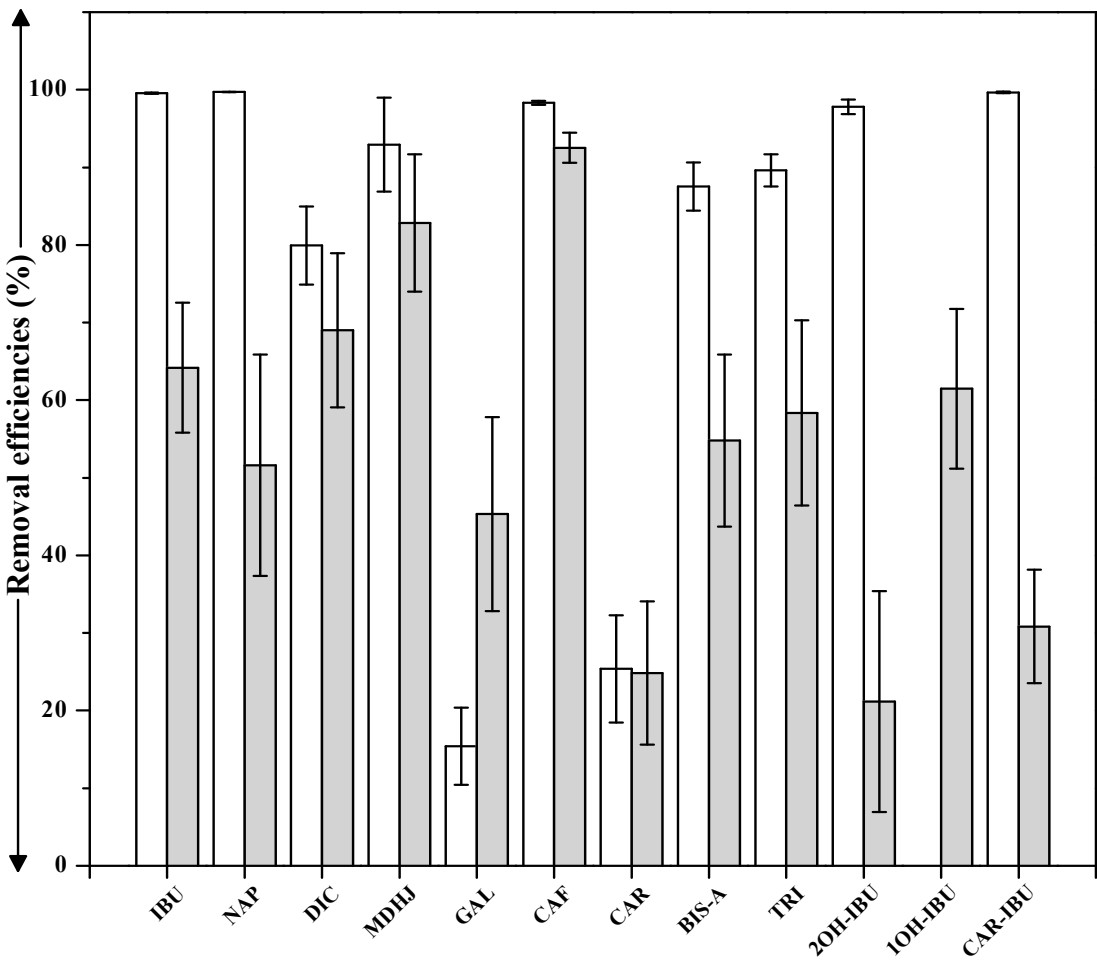

**Figure 4.** Removal efficiencies (bar chart) of OMPs for AS ( ☐ ) and HSSFs ( ▨ ). (Abbreviations used: ibuprofen (IBU); naproxen (NAP); diclofenac (DIC); methyl dihydrojasmonate (MDHJ); galaxolide (GAL); caffeine (CAF); carbamazepine (CAR); bisphenol-A (BIS-A); triclosan (TRI); 2-hydroxy ibuprofen (2OH-IBU); 1-hydroxy ibuprofen (1OH-IBU) and carboxy-ibuprofen (CAR-IBU)). Error bars represent standard deviations.

In the case of diclofenac, this compound was effectively removed by both systems (AS and HSSFs) with average percentages above 70% ($p > 0.05$). An opposite result was observed by Matamoros et al. [20] which showed a difference of 22% between AS and HSSFs in warm season. However, it is important to mention that these removal efficiencies achieved in this study were highly variable and ranged between 0%–90%. These results are in agreement with Ávila et al. [46] and Clara et al. [47]. Negative removal efficiencies (below 0%) may appear due to the conversion of their glucuronides and other transformations products to their parent compound through enzymatic processes in the WWTPs [48].

Regarding carbamazepine removal, the efficiencies achieved were 25% and 23% for AS and HSSFs, respectively without significant difference ($p > 0.05$). It is well known that carbamazepine is a recalcitrant compound that is not easily removed by conventional AS [44,47] or HSSF systems [22,45]. The recalcitrance of carbamazepine is due to heterocyclic structures such as pyridine. Moreover, uracils and furans are resistant to biodegradation under anaerobic conditions (ORPs below −70 mV) [49]. Park et al. [50] indicated that different CW configuration systems rarely remove more than 60% of carbamazepine. However, Dordio et al. [4] observed higher removal efficiencies (88% in winter and 97% in summer) for a microcosm-scale CW planted with *Typha* spp., where clay was used as a substrate, suggesting that carbamazepine can be removed mainly through sorption.

As expected, the removal of caffeine was above 90% in both systems without significant difference between AS and HSSFs. Other studies have indicated that this pharmaceutical is easily removed by WWTPs (higher than 90%) [5,47] and CWs [18,22]. Moreover, caffeine biodegradation is due to the hydrophilic properties (log Kow: 0.16) (Table 1) and is easily degraded by enzymatic complexes of microorganisms by demethylation [51]. The same tendency was observed for methyl dihydrojasmonate in which removal efficiencies were above 85% without a significant difference between AS and HSSFs. The structural simplicity of this molecule indicates that it can be removed by biodegradation (Table 1). By contrast, the lipophilic fragrance galaxolide (log Kow: 5.90) is removed mainly by sorption, which can occur in WWTPs or CWs [22,45]. In fact, in this study, galaxolide was determined in the particulate and dissolved fractions, AS and HSSF biosolids (Table 5). The removal efficiency of galaxolide from the dissolved fraction was 15% for AS and 50 % for HSSFs.

On the other hand, the removal efficiency of bisphenol-A in AS reached 87%. This is in agreement with Staples et al. [52], who found that bisphenol-A is easily biodegraded by WWTPs (half-live of 2.5 to 4 days). Several authors reported that up to 90% of bisphenol-A is removed by conventional WWTPs [47,53]. In contrast, the HSSF system was able to remove only 55% of bisphenol-A due to the system's low ORP (below −241 mV). Moreover, Ávila et al. [46] showed 65% removal of bisphenol-A under anaerobic conditions. They also showed that the biodegradation of bisphenol-A can be improved under oxidizing conditions in CWs. In addition, Navia et al. [54] found that compounds with phenols structure can be removed by sorption.

In the case of triclosan, the removal of this compound was 30% higher in the AS than in the HSSFs ($p < 0.05$). Similar results were observed in Matamoros et al. [22] where the removal of these compounds in AS system achieved 59% higher than in HSSF system. Several authors suggested that the triclosan removal mechanism by WWTPs is adsorption under aerobic conditions at efficiencies that ranged from 66% to 82% [36,55,56].

Regarding the transformation products, over 97% of carboxy-ibuprofen, 1-hydroxy ibuprofen, and 2-hydroxy ibuprofen was removed by AS. In contrast, the HSSF system had less effective removal percentages of 29%, 67%, and 25% for carboxy-ibuprofen, 1-hydroxy ibuprofen, and 2-hydroxy ibuprofen, respectively. Ibuprofen and its transformation products are easily removed by AS and HSSFs, but their removal efficiencies depend on operating conditions and system design [35]. Bendz et al. [57] showed removals of 95% and exceeding 96% for hydroxy ibuprofen and carboxy ibuprofen, respectively, which may have been due to the operating conditions of the AS system. Ferrando-Climent et al. [39] found that ibuprofen, carboxy-ibuprofen, and 2-hydroxy ibuprofen were easily removed by WWTPs, but 1-hydroxy ibuprofen presented some recalcitrance to biodegradation.

The results of this study showed that the AS system achieved better performances for removing OMPs than CWs. In general, the removal efficiencies of OMPs in AS technology were above 70%, except for galaxolide and carbamazepine where this system achieved percentages below 30%. This difference between AS and HSSF systems was also observed by Qiang et al. [23]. In this case, the removal efficiencies of some OMPs in AS varied between 72%–99% whereas CWs presented performances between 50%–79%. Moreover, Matamoros et al. [22] showed the same tendency where the average removal efficiency of OMPs in AS was 1.5 times higher than CWs. Although AS technology is effective for removing these types of compounds, one disadvantage is related to temperature. Some studies revealed that during warm conditions, the removal efficiencies decreased by up to 50% [22,23,58]. Moreover, the comparison of different wastewater treatment technologies should not only consider the performances achieved by different systems. It is important to determine the population that will use this technology. For rural areas, AS is not a good alternative because this technology has a high cost of construction and its operation needs trained professionals. CWs are a feasible option due to their better landscape integration and their low maintenance costs [59]. In this study, HSSF reported the lowest overall removal efficiency with average removal efficiency of 55%. As these systems promote anaerobic conditions, some OMPs were not removed efficiently [60]. Unsaturated systems such as VSSFs that are operated under aerobic conditions, reported for galaxolide, methyl dihydrojasmonate, tonalide,

diclofenac, and ibuprofen removal efficiencies of 88%–99%, 78%–99%, 75%–82%, 57%–73%, and 55%–99%, respectively [61–63]. For future studies, it is necessary to consider other configurations such as VSSFs or hybrid systems that ensure higher performance. Also, OMPs removal will be important to consider tertiary treatment via physical processes [9,64] or advanced oxidation processes [65] can reduce the recalcitrant compounds by adsorption and destruction, respectively.

## 4. Conclusions

Conflicting results have been reported when comparing the performance of conventional and non-conventional technologies on the OMPs removal that are presented in wastewater. In the case of this study, the results showed that the AS system had the best removal performance of contaminants. Regarding water quality parameters, the removal efficiencies of organic matter and nutrients in AS varied between 74%–99% whereas in HSSFs, these values fluctuated between 35%–80%. The same tendency was observed in the removal of OMPs where significant differences of 35%, 48%, 33%, 30%, 77%, and 69% were determined for naproxen, ibuprofen, bisphenol A, triclosan, 2-hydroxy ibuprofen, and carboxy ibuprofen between AS and HSSFs, respectively ($p < 0.05$). In the case of these compounds, the removal or biodegradation was performed more efficiently in aerobic conditions. Only for galaxolide, the removal efficiencies achieved by HSSF was three times higher than in AS ($p < 0.05$). Moreover, in both systems (AS and HSSF), carbamazepine, due to its chemical properties, was difficult to be removed with performance between 23%–25%. Despite the lower performances on OMPs removal achieved by HSSF, several modifications on the design and operational characteristics of these systems are necessary to ensure the use of non-conventional technologies in rural areas.

**Author Contributions:** Conceptualization, C.R.C. and G.V.; methodology, C.R.C. and J.M.B.; software, C.D.; validation, C.R.C., A.M.L., D.L. and G.V.; formal analysis, C.R.C., A.M.L., D.L.; investigation, C.R.C.; resources, J.M.B., G.V., C.R.C.; data curation, C.D.; writing—original draft preparation, C.R.C., D.L., A.M.L.

**Funding:** This research was funded by CONICYT, grants: FONDECYT Postdoctoral 3140162, CONICYT-PFCHA/ Doctorado Nacional/2019-21191116 and CONICYT/FONDAP/15130015.

**Acknowledgments:** This study was funded by the following grants: FONDECYT Postdoctoral 3140162 and CONICYT/FONDAP/15130015. A.M.L. thanks CONICYT for her Scholarship Program CONICYT-PFCHA/ Doctorado Nacional/2019-21191116, for supporting her Ph.D. studies at the University of Concepción.

**Conflicts of Interest:** The authors declare no conflict of interest.

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
