# Peer review of "Removal of Organic Micropollutants in Wastewater Treated by Activated Sludge and Constructed Wetlands: A Comparative Study"

_water, doi:10.3390/w11122515_

Round 1

Reviewer 1 Report

Authors present an interesting manuscript describing a quite straight forward study. I recommend its publication have some revisions

The comparison of Activate sludge treatment and CWs is not new as authors pointed out in the Introduction. Authors suggest that there are some contradictory findings in the literature. I assume that authors mean that in some cases AS is better than CWs and in others it is the opposite, but this is not clear in the Introduction because the last sentence before the aim is not clear. Authors also mention that CWs can have different design and in fact, CWs design is a major conditioner of removal performance and it is already know that horizontal CWs have low performance for organic micropollutants removal.

So, this should be more clear in the Introduction, more information regarding CWs design and indicate if there are other studies that also compared both technologies. More information on the advantages of one technology over the other should also be included.

In the aim it should be indicated which OMP were analysed.

In the experimental part it is mentioned that the AS is followed by a chlorination system. The effluent collected from AS is before or after chlorination? This is very important.

Authors mentioned “A total of 24 wastewater samples were obtained” How? 8 samples in triplicate? Only one sample at each time or three samples each sampling time? These details need clarification.

The effluents form the CW is a mixture from the effluents of the 4 CWs, this should be indicated in the text.

In the results and discussion it should be indicate if this comparison is in accordance with other studies or not. Moreover it should also be indicated the advantages and disadvantages of each technology. It should also be discussed that this design of CWs might not be the appropriate and maybe another CW design could be better.

Author Response

Answers in attached file

Reviewer 2 Report

The article Removal of organic micropollutants in wastewater treated by activated sludge and constructed wetlands: a comparative study is interesting and definitely suitable for publication in WATER.  However, I have several comments to the authors. I am positive that you will have no problems with improving the article. The justification of research presented in the Introduction chapter is insufficient. It is not enough to state that opinions on the removal of micropollutants are divided. Due to a wide range of interesting research, there should be no problem with expanding the justification.

General comments

Introduction

Line 58-59 Please describe what “conventional and non-conventional methods” are. Is constructed wetland a non-conventional method?

Is it possible to quote the possibility of applying constructed wetlands with vertical flow for micropollutants removal? SS VF systems are definitely more common in comparison with SS HF systems.

As mentioned above, conducting this research should be better (differently) justified. Lines 70-73 should be corrected (aim of the study should be extended).

Material and methods

My suggestion is to remove figure 1. The graphic representation in such form does not show the difference which occurs in case of AS (flow 2211 m3/d) and CW (flow only 0,088 m3/d) systems.

As I understood, the primary (preliminary in fig. 1) treatment was different in AS and C.W.?

Why were three zones applied in the CW system – the total length of CW was only 3 meters. On the other side, table 5 shows some interesting differences in results during CW treatment.

What does a horizontal line mean in Figure 2 (inside a CW bed)?

Please describe clearly what sewage was analyzed in the research. In lines 194-195 you wrote „the influent in this study can be classified as diluted wastewater”.

Were micropollutants presented in table 3 added to raw sewage or their concentration was natural in the analyzed sewage? It should be described clearly in the Material and Methods chapter.

Table 3 should be graphically modified: labels such as  pKa, Log Kow and Log Koc should be added. The column „Molecular structure” can be removed.

Results and discussion

Table 2: There is no such definition as “average pH”. Please leave only the range of pH.

Conclusions

I suggest extending conclusions due to a wide range of your research.

Please give some more results of micropollutants removal (AS versus CW system).

Author Response

Answers in attcahed file

Reviewer 3 Report

Water-634886 - Removal of organic micropollutants in wastewater treated by activated sludge and constructed wetlands: a comparative study. Reviewer’s comments.

Methods:

The area and depth of each HSSF bed is given (4.5m) but not their length and width, which is of interest in relation to placement of sampling pipes. The water depth is given twice (0.4m). Refer to Fig. 2 in 86-92 for clarity and brevity.

The wording of sampling strategy for WW is not clear (99-102); (Omit the word “two” in 100). The authors must mean that inlet samples were taken at a single point (not separate inlet samples for both AS and HSSF systems). Hence there are 3 sampling points (on each of 4 occasions) 1. inlet (after primary treatment), 2. AS effluent, 3. HSSF combined effluent. Hence, a total of 24 samples (2 x 4 x 3 = 24). Also, it should be stated that the HSSF effluent samples were from the combined beds (4 beds; 2 plant species). Is this right?

From Fig. 2, The CW bed does not appear to be a true horizontal flow with water level maintained at 0.4m throughout, since the outflow pipe is at base level. This would result in free-drainage, with water level declining from inlet to outlet, and aerobic conditions in much of the bed volume. Is there an external standpipe to maintain the internal water level? If not, the bed will not behave as HSSF – please explain.

Note, re 213 - Anaerobic denitrification is known to be important in HSSF systems, given adequate available carbon source and residence time.

Results:

Fig. 3 What are the error bars shown here? (std dev +/-; 95% conf int)? Similarly, in Fig 4, what are the error bars?

It is not clear how statistical comparisons between AS and HSSF systems have been made (175-178) - what values have been compared? Wilcoxon non-parametric test compares median values, whereas data Table 2 shows means and ranges for each measured parameter, and Tables 3-5 show means +/- SD. For the parameters in Table 2, it is not clear whether comparisons were made separately between the medians of inlet/effluent for AS, and inlet/effluent for HSSF, or whether the AS/HSSF values were compared directly. What comparisons do the P-values refer to? Wilcoxon’s test is used for “paired sample analysis” whereby differences matched paired values for inlet and effluent samples taken on the same day can be compared. Otherwise, week-to-week variation would influence the range of values comprising each mean/median, invalidating the test. It is not clear whether the analysis has been done on paired samples, such that the differences between AS and HSSF treatments were compared. Clearly, the 4 repeated measures of each parameter are not independent statistical replicates of a mean – they comprise a time series - hence the mean/median inlet should not me compared with mean/median effluent, unless the values are paired, as argued above. How was the analysis done?

224 should be Table 3, not 2. In the case of OMPs, statistical comparisons are made on removal % values, hence the paired sample analysis (paring between inlet and effluent for each system) is taken into account. The methods section should explain clearly that the samples were paired in the correct way (date-by-date), if that was indeed the case.

There are numerous minor errors of English, e.g. redundant use of “the” (32, 100, 186); omission of “as” in “such as…” (40, 41, 202); inappropriate verb tense “have been…” (33, 58); singular/plural forms - was/were (46, 107) samples (106); unclear sentence (68/69); consists of (80); “preliminary” (Fig.1); change “of” to “for” (187), “possibly” (198).

Author Response

Anwers in attached file

Round 2

Reviewer 3 Report

The revised m/s is greatly improved and clarification is provided on methods and presentation. 

A few minor errors have been introduced in the revised text:-

55. delete "a" (sentence uses plural form)

85. delete "to" (redundant word)

126. delete "two" (redundant word - duplicates "twice")

405. consider

407. ...and its operation needs trained...

413/4. ...necessary to consider...

416. Instead of "Divided", "Conflicting" would be a better word here.

Author Response

Dear Editor,

Attached you cna find the response to the reviewer´s comments.

With my best regards,

Gladys
